# Efficient Federated Learning on Knowledge Graphs via Privacy-preserving Relation Embedding Aggregation

**Kai Zhang[1], Yu Wang[2], Hongyi Wang[3], Lifu Huang[4], Carl Yang[5], Lichao Sun[1]**
[1]Lehigh University, [2]University of Illinois Chicago
[3]Carnegie Mellon University, [4]Virginia Tech, [5]Emory University
kaz321@lehigh.edu

## Abstract

Federated Learning (FL) on knowledge graphs (KGs) has yet to be as well studied as other domains, such as computer vision and natural language processing. A recent study FedE first proposes an FL framework that shares entity embeddings of KGs across all clients. However, compared with model sharing in vanilla FL, entity embedding sharing from FedE would incur severe privacy leakage. Specifically, the known entity embedding can be used to infer whether a specific relation between two entities exists in a private client. In this paper, we first develop a novel attack that aims to recover the original data based on embedding information, which is further used to evaluate the vulnerabilities of FedE. Furthermore, we propose a **Fed**erated learning paradigm with privacy-preserving **R**elation embedding aggregation (FEDR) to tackle the privacy issue in FedE. Compared to entity embedding sharing, relation embedding sharing policy can significantly reduce the communication cost due to its smaller size of queries. We conduct extensive experiments to evaluate FEDR with five different embedding learning models and three benchmark KG datasets. Compared to FedE, FEDR achieves similar utility and significant (nearly $2\times$) improvements in both privacy and efficiency on link prediction task.

## 1 Introduction

Knowledge graphs (KGs) are critical data structures to represent human knowledge, and serve as resources for various real-world applications, such as recommendation (Gong et al., 2021), question answering (Liu et al., 2018), disease diagnosis (Chai, 2020), etc. However, most KGs are usually incomplete and naturally distributed to different clients. Despite each client can explore the missing links with their own KGs by knowledge graph embedding (KGE) models (Lin et al., 2015),

---

Codes: https://github.com/taokz/FedR

Figure 1: FedE aggregates entity embeddings from clients while FEDR aggregates relation embeddings. Since in FEDR, there would be infinite embedding pairs of head and tail given a relation embedding, the inference attack would fail.

exchanging knowledge with others can further enhance completion performance because the overlapping elements are usually involved in different KGs (Chen et al., 2021; Peng et al., 2021).

Federated Learning (FL) allows different clients to collaboratively learn a global model without sharing their local data (McMahan et al., 2017). The first FL framework for KG – FedE is recently proposed, which learns the global-view and shared entity embeddings without collecting entire local KGs from different clients to a center server. However, at the very beginning in FedE, the server will collect the entity sets of every client for entity alignment (Chen et al., 2021), which will lead to unintentional privacy leakage.

**Data leakage problem in FedE.** Since the server maintains a complete table of entity embeddings with the user IDs, it could easily (1) identify client's users and (2) infer the relation embeddings by using the scoring function in KGE models, which could be defined as $f(h, r, t) \leq 0$ for all triples $(h, r, t)$. The details of scoring function are described in Table 7. As shown in Figure 1, for the server in FedE, the relation embeddings on each client can be obtained by calculating $r' = \arg\max_r f(h, r, t)$.

Since the aligned local head and tail entity embeddings in different clients are identical with each other after aggregation. Once the server can access the name of entities and relations by colluding with a single client, the local data of any target client including entities, relations, and the corresponding embeddings will be exposed.

To tackle the privacy issue in FedE, we propose FEDR based on relation embedding aggregation as illustrated in Figure 1. In FEDR, it would be impossible for the server to infer local entity embeddings given only relation embeddings. For example, we can not calculate $t' = \arg\max_t f(h, r, t)$ merely based on known $r$ but without $h$. Besides, the number of entities is usually much greater than the number of relations in real-world graph databases, so sharing relation embedding is much more communication-efficient.

We summarize the following contributions of our work. 1) We present a KG reconstruction attack method and reveal that FedE suffers a potential privacy leakage due to a malicious server and its colluded clients. 2) We propose FEDR, an efficient and privacy-preserving FL framework on KGs. Experimental results on three benchmark datasets demonstrate that FEDR has the competitive performance compared with FedE, but gains nearly $2\times$ improvements in terms of privacy protection and communication efficiency.

## 2 Methodology

### 2.1 Knowledge Graph Reconstruction

The purpose of this attack is to recover original entities and relations in a KG given traitor's information including parital or all triples and the corresponding embeddings, namely element-embedding pairs. We summarize the method into 4 steps:
(1) The server colludes with one client C1 to obtain its element-embedding pairs $\langle (E, e), (R, r) \rangle$.
(2) Infer the target client's element embedding such as relation embedding by calculating $r' = \arg\max_r f(h, r, t)$ where $h, r \in e$.
(3) Measure the discrepancy between the inferred element embedding such as relation embedding $r'$ and all known $r$ with cosine similarity.
(4) Infer the relation $R'$ as $R$, $E'$ as $E$ with corresponding largest similarity scores.

The whole attack progress in different cases are included in Appendix B. Note that in the step (1) mentioned above, the server could also collude

| LR | 30% | | 50% | | 100% | |
|---|---|---|---|---|---|---|
| | ERR | TRR | ERR | TRR | ERR | TRR |
| C1 | 0.3000 | 0.0647 | 0.5000 | 0.2045 | 1.0000 | 0.7682 |
| C2 | 0.2904 | 0.0607 | 0.4835 | 0.1951 | 0.9690 | 0.7378 |
| C3 | 0.2906 | 0.0616 | 0.4846 | 0.1956 | 0.9685 | 0.7390 |

Table 1: Privacy leakage on FB15k-237 with TransE.

with any malicious clients.

**Privacy leakage quantization in FedE.** We define two metrics: *Triple Reconstruction Rate* (TRR) and *Entity Reconstruction Rate* (ERR). TRR measures the ratio of correctly reconstructed triples by inferring relations between two entities as described above. ERR measures the ratio of entities that the server can reveal their names to the whole number of entities. We let the server owns 30%, 50%, 100% trained element-embedding pairs from C1, the traitor, to reconstruct entities and triples of others. In FedE, ERR simply reflects the portion of entities that C1 shares with the server. The results of privacy leakage on FB15k-237 (Toutanova et al., 2015) over three clients are summarized in Table 1. LR in tables denotes information (entities, the corresponding relations and relation embeddings) leakage ratio from C1. It is clear that the server only needs to collude with one client to obtain most of the information of KGs on other clients. In a word, FedE is not privacy-preserving.

### 2.2 FEDR

Compared to single-silo learning, FEDR and FedE learn better representations by taking advantage of the complementary capabilities from cross-clients information. To protect the data privacy in FL on KGs, FEDR adopts two strategies: (1) Before aggregation works, the server acquires all IDs of the unique relations from local clients and maintains a relation table via Private Set Union (PSU), which computes the union of relations, without revealing anything else, for relation alignment (Kolesnikov et al., 2019). Therefore, although the server still maintains the relation table, the server does not know the relations each client holds. (2) Different from sharing entity embeddings, in FEDR, each client first trains its own entity and relation embeddings locally, and only sends relation embeddings to the server. The server will aggregate the aligned relation embeddings and dispense them to clients for further local updates. The client-side embedding training depends on the type of local KGE models such as translation distance models and se-

mantic matching models (Sun et al., 2020). More details are described in Appendix A.

**Privacy Enhancement.** Although the relation privacy could be achieved by PSU, the server still can roughly infer the relation by comparing the uploaded relation embedding with the one stored in the relation table. Therefore, to further guarantee no raw data leakage, Secure Aggregation (Bonawitz et al., 2017) is exploited to protect the privacy of any individual relation embeddings. The fundamental idea behind it is to mask the uploaded embeddings such that the server cannot obtain the actual ones from each client. However, the sum of masks can be canceled out, so we still have the correct aggregation results.

## 3 Experiments

We carry out several experiments to explore FEDR's performance in link prediction, in which the tail $t$ is predicted given head $h$ and relation $r$.

**Datasets.** We evaluate our framework through experiments on three public datasets, FB15k-237, WN18RR (Dettmers et al., 2018) and a disease database – DDB14 (Wang et al., 2021). To build federated datasets, we randomly split triples to each client without replacement, then divide the local triples into the train, valid, and test sets with a ratio of 80/10/10. The statistics of datasets after split is described in Table 2.

**KGE Algorithms.** Four commonly-used KGE algorithms – TransE (Bordes et al., 2013), RotatE (Sun et al., 2019), DisMult (Yang et al., 2014) and ComplEx (Trouillon et al., 2016) are utilized in the paper. We also implement federated NoGE (Nguyen et al., 2022), a GNN-based algorithm.

Note that, although random split makes data homogeneous among all the clients to some extent, it ensures fair comparison between FedE and FEDR. Otherwise, if we build subgraphs in terms of relation types using non-iid partition, FedE will definitely outperform FEDR because of less overlapping relations among clients.

### 3.1 Effectiveness Analysis

The commonly-used metric for link prediction, mean reciprocal rank (MRR), is exploited to evaluate FEDR's performance. We take FedE and Local, where embeddings are trained only on each client's local KG, as the baselines. Table 3 shows the link prediction results under settings of different number of clients $C$. We observe that

| Dataset | #C | #Entity | #Relation |
|---------|-----|---------|-----------|
| DDB14 | 5 | $4462.20_{\pm 1049.60}$ | $12.80_{\pm 0.84}$ |
| | 10 | $3182.60_{\pm 668.89}$ | $12.60_{\pm 0.70}$ |
| | 15 | $2533.86_{\pm 493.47}$ | $12.50_{\pm 0.74}$ |
| | 20 | $2115.59_{\pm 385.56}$ | $12.35_{\pm 0.75}$ |
| WN18RR | 5 | $21293.20_{\pm 63.11}$ | $11.00_{\pm 0.00}$ |
| | 10 | $13112.20_{\pm 46.70}$ | $11.00_{\pm 0.00}$ |
| | 15 | $9537.33_{\pm 45.45}$ | $11.00_{\pm 0.00}$ |
| | 20 | $7501.65_{\pm 31.72}$ | $11.00_{\pm 0.00}$ |
| FB15k-237 | 5 | $13359.20_{\pm 27.36}$ | $237.00_{\pm 0.00}$ |
| | 10 | $11913.00_{\pm 31.56}$ | $237.00_{\pm 0.00}$ |
| | 15 | $10705.87_{\pm 36.93}$ | $236.87_{\pm 0.35}$ |
| | 20 | $9705.95_{\pm 44.10}$ | $236.80_{\pm 0.41}$ |

Table 2: Statistics of federated datasets. The subscripts denote standard deviation. # denotes "number of".

FEDR comprehensively surpasses Local under all settings of the number of clients, which indicates that relation aggregation makes sense for learning better embeddings in FL. Take NoGE as an example, FEDR gains $29.64 \pm 0.037\%$, $22.13 \pm 0.065\%$, and $11.84 \pm 0.051\%$ average improvement in MRR on three dataset. Compared with FedE, FEDR usually has the better or similar results with the KGE models of DistMult and its extensive version ComplEx on all datasets. We also observe that FedE fails to beat Local setting and even performs catastrophically with these two KGE models on both DDB14 and WN18RR. Although FedE performs better than FEDR with TranE and RotatE, the absolute performance reductions between FedE and FEDR are mostly (13/16 = 81%) within 0.03 in MRR on both DDB14 and FB15k-237, which illustrates that FEDR is still effective. The theoretical explanations behind these results *w.r.t* data heterogeneity, and characteristics of FL and KGE models need further studies.

### 3.2 Privacy Leakage Analysis

Compared with entity aggregation, additional knowledge is required to make privacy leakage in FEDR because it is almost impossible to infer any entity or triple from relation embeddings only. Therefore, we assume the server can access part or all entity embeddings from clients. The information leakage ratio of local entity embeddings (LLR) set as $30\%, 50\%, 100\%$ respectively in the experiment. For simplicity, we let the server holds all entity embeddings from C1 in Section 2.1, i.e., LLR=100%. Besides, for fair comparison, any encryption techniques mentioned in Section 2 are not taken into account in this privacy analysis.

Figure 2 presents the privacy leakage quantiza-

| Dataset | | DDB14 | | | | WN18RR | | | | FB15k-237 | | | |
|---|---|---|---|---|---|---|---|---|---|---|---|---|---|
| Model | Setting | C = 5 | C = 10 | C = 15 | C = 20 | C = 5 | C = 10 | C = 15 | C = 20 | C = 5 | C = 10 | C = 15 | C = 20 |
| TransE | Local | 0.4206 | 0.2998 | 0.2464 | 0.2043 | 0.0655 | 0.0319 | 0.0378 | 0.0285 | 0.2174 | 0.1255 | 0.1087 | 0.0874 |
| | FedE | 0.4572 | 0.3493 | 0.3076 | 0.2962 | 0.1359 | 0.1263 | 0.1204 | 0.1419 | 0.2588 | 0.2230 | 0.2065 | 0.1892 |
| | FEDR | **0.4461** | 0.3289 | 0.2842 | 0.2761 | 0.0859 | 0.0779 | 0.0722 | 0.0668 | **0.2520** | 0.2052 | 0.1867 | 0.1701 |
| RotatE | Local | 0.4187 | 0.2842 | 0.2411 | 0.2020 | 0.1201 | 0.0649 | 0.0513 | 0.0155 | 0.2424 | 0.1991 | 0.1526 | 0.0860 |
| | FedE | 0.4667 | 0.3635 | 0.3244 | 0.3031 | 0.2741 | 0.1936 | 0.1287 | 0.0902 | 0.2682 | 0.2278 | 0.2199 | 0.1827 |
| | FEDR | 0.4477 | 0.3184 | 0.2765 | 0.2681 | 0.1372 | 0.1271 | 0.1074 | **0.0912** | 0.2510 | 0.2080 | 0.1854 | 0.1586 |
| DistMult | Local | 0.3037 | 0.2485 | 0.2315 | 0.1877 | 0.1137 | 0.0946 | 0.0766 | 0.0670 | 0.1133 | 0.0773 | 0.0765 | 0.0689 |
| | FedE | 0.2248 | 0.1145 | 0.0764 | 0.0652 | 0.0654 | 0.0517 | 0.0548 | 0.0374 | 0.1718 | 0.1129 | 0.0901 | 0.0753 |
| | FEDR | **0.4219** | **0.3146** | **0.2685** | **0.2577** | **0.1350** | **0.1202** | **0.1198** | **0.0898** | **0.1670** | 0.0999 | **0.0884** | **0.0814** |
| ComplEx | Local | 0.3595 | 0.2838 | 0.2411 | 0.1946 | 0.0153 | 0.0115 | 0.0108 | 0.0122 | 0.1241 | 0.0694 | 0.0571 | 0.0541 |
| | FedE | 0.3406 | 0.2025 | 0.1506 | 0.1247 | 0.0035 | 0.0013 | 0.0003 | 0.0022 | 0.1603 | 0.1161 | 0.0944 | 0.0751 |
| | FEDR | **0.4287** | **0.3235** | **0.2747** | **0.2611** | **0.0203** | **0.0152** | **0.0152** | **0.0166** | **0.1716** | **0.1174** | **0.1075** | **0.0993** |
| NoGE | Local | 0.3178 | 0.2298 | 0.1822 | 0.1580 | 0.0534 | 0.0474 | 0.0371 | 0.0372 | 0.2315 | 0.1642 | 0.1246 | 0.1042 |
| | FedE | 0.3193 | 0.3171 | 0.2678 | 0.2659 | 0.0789 | 0.0697 | 0.0632 | 0.0533 | 0.2412 | 0.1954 | 0.1730 | 0.1637 |
| | FEDR | **0.4312** | **0.3127** | **0.2604** | 0.2452 | 0.0669 | 0.0543 | 0.0530 | 0.0499 | **0.2432** | 0.1822 | 0.1448 | 0.1282 |

Table 3: Link prediction results (MRR) with $C = 5, 10, 15$ and $20$. **Bold** number denotes FEDR performs better than or close to (within 3% performance decrease) FedE. Underline number denotes the better result between FEDR and Local.

| | GEE | LEE | GRE | LRE |
|---|---|---|---|---|
| FedE | ✔ | ✔ | ✘ | ✘ |
| FedR | ✘ | ✔ | ✔ | ✔ |

Table 4: Summary of adversary knowledge in knowledge graph reconstruction attack. "G" represents "Global", "L" represents "Local". "EE" and "RE" represent entity and relation embeddings, respectively.

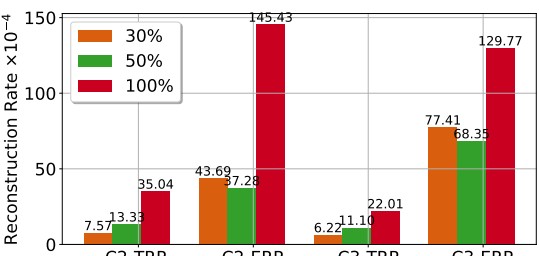

Figure 2: Privacy leakage in FEDR on FB15k-237.

| Dataset | WN18RR | | DDB14 | |
|---|---|---|---|---|
| | ERR | TRR | ERR | TRR |
| C2 | 22.00 | 9.89 | 19.39 | 10.10 |
| C3 | 18.44 | 9.23 | 8.87 | 5.05 |

Table 5: Privacy leakage in FEDR on WN18RR and DDB14 ($\times 10^{-4}$) when LLR = 100%.

tion in FEDR over three clients on FB15k237, in which the scale of the Y-axis is in $10^{-4}$. The results demonstrate that relation aggregation can guarantee both entity-level and graph-level privacy even if providing additional local entity embeddings. In addition, we summarize the difference of adversary knowledge in FedE and FEDR in Table 4. We observe that despite the relation embedding can be exploited directly in FEDR instead of inference, the privacy leakage rates in FEDR are still substantially lower than the ones in FedE. For example, according to Table 1, for C2, FEDR obtains reduction $(9690 - 145.43 = 9544.57) \times 10^{-4}$ and $(7378 - 35.04 = 7342.96) \times 10^{-4}$ in ERR and TRR (which is about 98.50% and 99.52% relative reduction) on FB15k237, respectively. To explain the result intuitively, in FEDR, local entity embeddings of the same entity in each client usually vary. Therefore, calculating the similarity between embeddings to reconstruct KGs does not work.

Results of privacy leakage quantization (LLR = 100%) on other datasets are shown in Table 5. We observe that both ERRs and TRRs on two datasets are very low, which is consistent with the results on FB15k-237. Therefore, FEDR can successfully defense against KG reconstruction attack and gain small privacy leakage rates.

### 3.3 Communication Efficiency Analysis

In this section, the product of data sizes and communication rounds is calculated to measure the communication cost. Considering the performance difference between FEDR and FedE, for fair comparison of communication efficiency, we count the communication rounds when the model reaches a pre-defined MRR target on the validation dataset, specifically, we set two different MRR targets: 0.2 and 0.4. Since all models perform well on DDB14, we take the setting with $C = 5$ on DDB14 as an example in this section. The required communication rounds for each model are depicted in Figure 3. We observe that FEDR reaches the target with much less communication rounds compared with FedE. For instance, FEDR-DistMult reaches

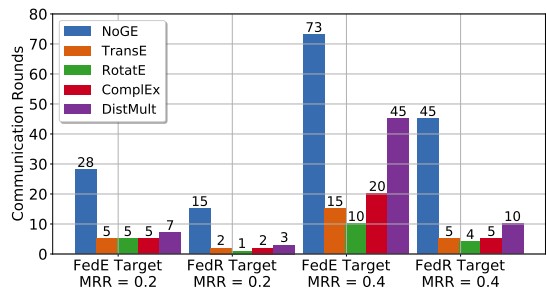

Figure 3: Number of communication rounds to reach a target MRR for FedE and FEDR with a fixed $C = 5$.

| $\epsilon$ | 0.5 | 1 | 5 | 10 | w/o |
|---|---|---|---|---|---|
| MRR | 0.0035 | 0.0072 | 0.0589 | 0.0693 | **0.5954** |
| TRR | 0.0412 | 0.0421 | 0.0490 | 0.0521 | **0.5091** |

Table 6: Experimental results with and without DP.

the target MRR = 0.4 within 10 communication rounds while FedE uses 45 rounds. Also, according to statistics of federated datasets in Table 2, the average of the number of unique entities in FedE and unique relations in FEDR are 4462.2 and 12.8, respectively. We ignore the embedding size and just use the number of entities/relations to reflect data size. By using relation aggregation, $99.89 \pm 0.029\%$ of communication cost is reduced in average for all clients when the target MRR is 0.2, while $99.90 \pm 0.042\%$ of communication cost is reduced in average when the target MRR is 0.4. These results demonstrate that our proposed framework is much more communication-efficient.

## 4 Discussion

One potential concern will be discussed: can differential privacy (DP) used against KG reconstruction attack? To answer this question, we employ the vanilla local Laplace DP to mask the uploaded entity embeddings in FedE, where the sensitivity is set as 2 because the value of components in an embedding ranges from -1 to 1.

In this experiment, we evaluate FedE-DP on DDB14 dataset with TransE KGE model while we set LR = 100%. The experimental results of C2 *w.r.t* MRR and TRR based on different privacy budgets $\epsilon$ are shown in Table 6. We can see that the TRRs drop from 0.5091 to approximately 0.04 when applying DP. However, although DP can defend the KG reconstruction, it also degrades the model performance on the link prediction task, where the MRR drops from 0.5954 to less than 0.1. In this case, we can conclude that vanilla local Laplace DP is not a effective solution for reconstruction attack in FedE.

Besides, there are some interesting but unsolved problems left in our work, which provide future research opportunities: 1) What's the explicit or implicit relations among data heterogeneity, aggre-

gation stretegies, and KGE methods? 2) Could the success rate of the KG reconstruction attack transfer to the privacy level of any formal privacy guarantee such as differential privacy?

## 5 Conclusion

In this paper, we conduct the first empirical quantization of privacy leakage to federated learning on knowledge graphs, which reveals that recent work FedE is susceptible to reconstruction attack based on shared element-embedding pairs when there are dishonest server and clients. Then we propose FEDR, a privacy-preserving FL framework on KGs with relation embedding aggregation that defenses against reconstruction attack effectively. Experimental results show that FEDR outperforms FedE w.r.t data privacy and communication efficiency but also keeps similar utility.

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

## A  Federated Knowledge Graph Embedding Framework

The federated knowledge graph embedding (FKGE) framework consists of two processes in one communication round: 1) server-side aggregation and 2) client-side update, which are summarized in Algorithm 1, where $\{\mathbf{P}^c\}$ denotes the set of permutation matrices and $\{\mathbf{v}^c\}$ denotes existence vectors. More specifically, $\mathbf{P}^c_{i,j} = 1$ indicates that the $i$-th element in element table is aligned with the $j$-th element of client $c$, while $\mathbf{v}^c_i = 1$ shows that the $i$-th element in element table exists in client $c$. Besides, $\oslash$ is element-wide division, $\otimes$ is element-wide multiplication, and $\mathbb{1}$ is an all-one vector.

---

**Algorithm 1:** A summary of FKGE that uses conventional KGE models and GNN-based KGE methods.

**Input** : local datasets $T^c$, number of clients $C$, number of local epochs $E$, learning rate $\eta$

**Server excutes:**
1   initialize $\mathbf{E}^*_0$
    * denotes $r$ – use FEDR or $e$ – use FEDE
2   **for** round $t = 0, 1, ...$ **do**
3     Sample a set of clients $C_t$
4     **for** $c \in C_t$ **do in parallel**
5       Send permuted relation embedding
6       matrix to client $c : \mathbf{E}^{*,c}_t \leftarrow \mathbf{P}^c\mathbf{E}_t$
7       $\mathbf{E}^{*,c}_{t+1} \leftarrow \text{Update}(c, \mathbf{E}_t)$
8     $\mathbf{E}^*_{t+1} \leftarrow (\mathbb{1} \oslash \sum_{c=1}^{C_t} \mathbf{v}^c) \otimes \sum_{c=1}^{C_t} \mathbf{P}^c\mathbf{E}^{*,c}_{t+1}$
9   **return** $\mathbf{E}^*$

**Client excutes** $\text{Update}(c, \mathbf{E})$:
10   **for** each local epoch $e = 1, 2, ..., E$ **do**
11     **for** each batch $\mathbf{b} = (\mathbf{h}, \mathbf{r}, \mathbf{t})$ of $T^c$ **do**
12       i) $\mathbf{E} \leftarrow \mathbf{E} - \eta\nabla\mathcal{L}$
13       ii) $w \leftarrow w - \eta\nabla\mathcal{L}$

    only one of i) and ii) will be implemented
14   **return** $\mathbf{E}^{*,c} \in \mathbf{E} := \{\mathbf{E}^{e,c}, \mathbf{E}^{r,c}\}$

---

## B  Knowledge Graph Reconstruction

We summarize the knowledge graph reconstruction attack in Algorithm 2. Note that in the algorithm, i) and ii) refer to different operations, and only one will be performed in FedE or FEDR.

## C  Implementation Details

For TransE, RotatE, DistMult, and ComplEx, we follow the same setting as FedE (Chen et al., 2021). Specifically, the number of negative sampling, margin $\gamma$ and the negative sampling temperature $\alpha$ are set as 256, 10 and 1, respectively. Note that, we adopt a more conservative strategy for embedding aggregation where local non-existent entities

will not be taken as negative samples compared to FedE. For NoGE, we use GCN (Kipf and Welling, 2016) as encoder and QuatE (Zhang et al., 2019) as decoder. Once local training is done in a communciation round, the embeddings are aggregated and the triplet is scored by the decoder. The hidden size of 1 hidden layer in NoGE is 128.

---

**Algorithm 2:** Knowledge graph reconstruction including attack in FEDE/FEDR.

**Adversary knowledge:** Local entity embeddings – **LEE**, local relation embeddings – **LRE**, element-embedding paris from a client – **EEP**, type of the used KGE model.

**Entity reconstruction:**
1   **for** entity embedding $\hat{e} \in \mathbf{LEE}$ **do**
2     **for** entity-embedding $(E, e) \in \mathbf{EEP}$ **do**
3       Calculate similarity between $e$ and $\hat{e}$
4       Update the inferred entity $\hat{E} = E$ with the greatest similarity score
5   **return** the reconstructed entity set $\{\hat{E}\}$

**Triple reconstruction:**
only one of i) and ii) will be implemented
6   i) **for** entity embeddings $(\hat{h}, \hat{t}) \in \mathbf{LEE}$ **do**
7     Calculate relation embedding $\hat{r}$ based on the scoring function of used KGE model, e.g. $\hat{r} = \hat{t} - \hat{h}$ with TransE
8     **for** relation-embedding $(R, r) \in \mathbf{EEP}$ **do**
9       Calculate similarity between $r$ and $\hat{r}$
10       Update the inferred relation $\hat{R} = R$ with the greatest similarity score
11   **return** the reconstructed relation set $\{\hat{R}\}$
12   ii) **for** relation embedding $\hat{r} \in \mathbf{LRE}$ **do**
13     **for** relation-embedding $(R, r) \in \mathbf{EEP}$ **do**
14       Calculate similarity between $r$ and $\hat{r}$
15       Update the inferred relation $\hat{R} = R$ with the greatest similarity score
16   **return** the reconstructed relation set $\{\hat{R}\}$
17   Utilize $\{\hat{E}\}$ and $\{\hat{R}\}$ to reconstruct triples.

---

Since the aggregated information is not exploited in the local training in NoGE, we also implement KB-GAT (Nathani et al., 2019), the other GNN model but it can take advantages of both graph structure learning and global-view information aggregation. However, Fed-KB-GAT is memory-consuming. For KB-GAT, we use GAT (Veličković et al., 2018) as encoder and ConvKB (Nguyen et al., 2017) as decoder. Although the input to KB-GAT is the triple embedding, this model update neural network weights to obtain the final entity and relation embeddings. In each communication, we let the aggregated embeddings be the new input to KB-GAT, we find using small local epoches lead to bad performance because the model is not fully trained

to produce high-quality embeddings. Therefore, we set local epoch of GAT layers as 500, while local epoch of convlutional layers as 150. Embedding size is 50 instead of 128 like others since we suffers memory problem using this model.

If not specified, the local update epoch is 3, the embedding dimension of entities and relation is 128. Early stopping is utilized in experiments. The patience, namely the number of epochs with no improvement in MRR on validation data after which training will be stopped, is set as 5. We use Adam with learning rate 0.001 for local model update.

## C.1 Scoring Function

| Model | Scoring Function |
|---|---|
| TransE | $-\|\mathbf{h}+\mathbf{r}-\mathbf{t}\|$ |
| RotatE | $-\|\mathbf{h}\circ\mathbf{r}-\mathbf{t}\|$ |
| DistMult | $\mathbf{h}^{\top}\operatorname{diag}(\mathbf{r})\mathbf{t}$ |
| ComplEx | $\operatorname{Re}\left(\mathbf{h}^{\top}\operatorname{diag}(\mathbf{r})\bar{\mathbf{t}}\right)$ |
| NoGE | $\langle a'_h, a_t\rangle + \langle b'_h, b_t\rangle + \langle c'_h, c_t\rangle + \langle d'_h, d_t\rangle$ |
| KB-GAT | $\left(\|_{m=1}^{\Omega} \operatorname{ReLU}\left(\left[\vec{h}_i, \vec{g}_k, \vec{h}_j\right] * \omega^m\right)\right) \cdot \mathbf{W}$ |

Table 7: A list of scoring functions for KGE models implemented in this paper. The scoring function used in NoGE comes from QuatE (Zhang et al., 2019).

# D Additional Results

## D.1 Convergence Analysis

The convergence curves considering four KGE models and three dataset are shown in Figure 4. The solid and dashed lines represent curves *w.r.t* FEDR and FedE, respectively. We do not show the curves of NoGE because the aggregated embeddings does not influence local training. We observe that FEDR usually converge faster than FedE.

## D.2 Experiment result with KB-GAT

We conduct KB-GAT with both entity aggregation and relation aggregation on DDB14 with $C = 3$

| Model | Setting | MRR | Hit@1 | Hit@3 | Hit@10 |
|---|---|---|---|---|---|
| RotatE | Local | 0.5347 | 0.5311 | 0.5459 | 0.5912 |
|  | FedE | 0.6087 | 0.5070 | 0.6774 | 0.7916 |
|  | FEDR | 0.5834 | 0.5583 | 0.5852 | 0.6326 |
| KB-GAT | Local | 0.5507 | 0.5361 | 0.5529 | 0.5754 |
|  | FedE | **0.7907** | **0.7366** | **0.7522** | **0.8650** |
|  | FEDR | 0.7501 | 0.7124 | 0.7620 | 0.8328 |

Table 8: Extensive experimental resutls on DDB14 with $C = 3$. **Bold** number denotes the best result in FedE and underline number denotes the best result in FEDR.

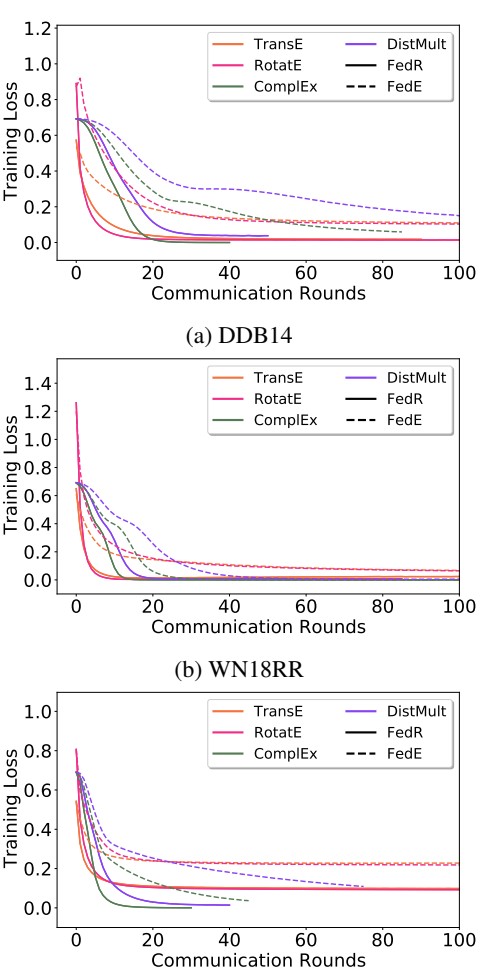

(a) DDB14

(b) WN18RR

(c) FB15k-237

Figure 4: Training loss versus communication ($C = 5$).

as shown in Table 8. Due to the good performance of RotatE, we also compare KB-GAT with RotatE. Hit@N is also utilized in the evaluation. From the table, KB-GAT beats RotatE in regard of all evaluation metrics in both FedE and FedR setting. However, how to implement federated KB-GAT in a memory-efficient way is still an open problem.