# OpenReview forum: "Efficient Federated Learning on Knowledge Graphs via Privacy-preserving Relation Embedding Aggregation"
_aclweb.org/ACL/2022/Workshop/FL4NLP — FL4NLP@ACL2022_

### Official Review · Reviewer_XHJY · 2022-03-14
**Review on "Efficient Federated Learning on knowledge Graphs via Privacy-Preserving Relation Embedding Aggregation"**

**Rating:** 7
**Confidence:** 3

**Review:**

This work presents an extension of FedE, a recently proposed knowledge graph aggregation scheme in Federated Learning. Specifically, the authors tackle the privacy issue by aggregating relation embedding instead of directly aggregating the entity embedding.

Although the idea is quite simple, it effectively addresses the privacy issue while maintaining the model performance. I appreciate this idea and advocate accepting the article in this workshop. I see a few things that can be improved as follows.

1. The experimental settings can be improved. The number of clients is obviously too small. Federated Learning is a large-scale distributed learning paradigm. Considering the total number of entities in the benchmark datasets, the dataset can be distributed to more than 20 clients. If the number of clients is hundreds for example, would similar performance benefits still be available?

2. Although FedE is the recently proposed representative work, comparing FedR to this single work does not provide useful insights. Additional comparisons to other works (especially referenced in FedE paper) will significantly strengthen the paper.

---

### Official Review · Reviewer_rKgA · 2022-03-23
**Good idea, but some inaccuracies**

**Rating:** 7
**Confidence:** 4

**Review:**

I think overall, this paper would be interesting for the workshop, and authors have proposed an interesting approach. There are issues in presentation and claims which can be improved and fixed. Hopefully, discussions at the workshop can help authors gather more feedback and continue their work.


Below are some comments which I hope authors will find useful to improve their work.

It seems that authors are proposing to modify the FL setting, by assuming that server might collude with the clients. This is a modified definition of privacy for FL which is fine and interesting. However, in the abstract and introduction, authors seem to claim that they have found a severe privacy leakage for FedE method and they want to address that severe shortcoming. Contribution of this paper can be explained more clearly starting with the assumptions used in FL literature, FedE method, and authors' method.

Statements like "FedE is not privacy-preserving" are meaningless without proper definition of "privacy". From authors' point of view, how many of the methods in FL literature can be considered "privacy-preserving"?

Phrases like "much more" for communication efficiency seem to be overly vague.

Function f(h,r,t) is not defined properly.

It is not clear what authors mean by "honest-but-curious" server. If the server is honest why should we go through all these trouble to hide the data? Authors have to define what they mean by "honest" and how that affects their formulation.

If a client is a traitor and shares the data with the server, why would it share only a percentage of its data and not all of it?

Method is not presented in a coherent way. First, it is mentioned that "To guarantee the data privacy in the FedE, FEDR adopts two main strategies". Despite this guarantee, a paragraph later, it is mentioned that "the server still can roughly infer the relation by comparing the uploaded relation embedding with the one stored...". It is not clear what "roughly" entails here and how it can combine with the statements before. It sounds like authors do not have a clear definition of privacy in mind, are they are putting bandaids on various shortcomings that they consider. The word "guarantee" seem to lose its meaning.

Figure 4 which provides the comparison with other methods does not depict the performance of proposed method. In the legend, proposed method is shown as a solid black line, but in the plot, there is no such line.

---

### Official Review · Reviewer_do3D · 2022-03-24

**Rating:** 6
**Confidence:** 3

**Review:**

This paper proposed a novel reconstruction attack method to infer the client's data utilizing the model updates in a knowledge graph setting. The proposed method achieves strong attacking accuracy on previous baselines. To overcome this, the authors further proposed a new method called FedR robust to the proposed reconstruction attack method.

The method is interesting. The idea of sharing relation embedding rather than head/tail embedding for aggregation is novel and seems to effectively prevent exact inference of the user data. The experimental results seem strong and significantly outperform prior work on multiple benchmarks. Here are some of my concerns:
- Rather than saying the proposed FedR is 'privacy-preserving', it seems that FedR is only robust to the proposed attacking method. There are a couple of things missing here: 1. How does the success rate of the attacking method transfer to the privacy level of the randomized algorithm to train KG? Could it transfer to any formal differential privacy guarantee? 2. How to evaluate the optimality of the attack evaluated in this work? Could there be stronger, defense-aware attack that could similarly break FedR? It would be great if the authors could provide additional details for these points.
- Could the authors add a related work/background section to list relevant work on reconstruction attacks/FL on KG?
- Could the authors add a clear algorithm description on FedR?
- In the description of experiment, the authors claim each dataset is randomly split among clients? Does that mean the data are homogeneous among all the clients? In a typical FL setting, data are heterogeneous on each client. Could the authors add experiments on experiments under heterogenous data?

---

### Official Review · Reviewer_r89r · 2022-03-24
**Review of the paper**

**Rating:** 6
**Confidence:** 3

**Review:**

This paper considers an important and timely problem in federated learning on knowledge graphs (KGs), especially developing an attack model which incurs privacy leakage in an existing work, named FedE, and proposing a new privacy-preserving embedding aggregation framework to protect the privacy against the attack.

This paper proposed the attack model which can reconstruct original entities and relations of individual client based on the local embedding matrix, which is an important finding in Federated Knowledge Graph Completion from the privacy perspective. It empirically demonstrates the effectiveness of the attack with a simple 3 clients-model. Then, this paper proposed a relation embedding aggregation framework to reduce the privacy leakage while it also reduces the required bandwidth to achieve the target MRR.

It is worth to discuss the proposed attack model and defense mechanism even though the reviewer has the following concerns.
1.	This paper does not contain the system model of FedE and attack model in the main body. Before reading the Appendix C and D, I cannot capture the which information is communicated between the server and clients, which information is known to the server and colluding client, and how to reconstruct the private local information. It would be helpful to provide the system model (including definition of entity/relation embeddings and  local/global update equation, and … etc)
2.	In addition to 1, it is unclear what is the global update equation based on the local relation embeddings in the proposed framework, FedR.
3.	There are minor typos such as: 1) there is (?) in Appendix A 2) paris => pairs in Algorithm 1 in Appendix C.

---

### Decision · Program_Chairs · 2022-03-26

Accept